# ESCA as a Tool for Exploration of Metals' Surface

**Eleonora Bolli** [1,2,*] , **Saulius Kaciulis** [2] **and Alessio Mezzi** [2]

1   Department of Industrial Engineering, University of Rome "Tor Vergata", Via del Politecnico 1,
    00133 Rome, Italy
2   Institute for the Study of Nanostructured Materials, ISMN—CNR, Monterotondo Stazione,
    00015 Rome, Italy; Saulius.kaciulis@cnr.it (S.K.); alessio.mezzi@cnr.it (A.M.)
*   Correspondence: Eleonora.bolli@ismn.cnr.it; Tel.: +39-06-90672892

**Abstract:** The main principles and development of electron spectroscopy for chemical analysis (ESCA) are briefly reviewed. The role of ESCA techniques (X-ray photoelectron spectroscopy and Auger electron spectroscopy) in the investigation of metallic surfaces is discussed, evidencing their importance and analytical potentiality. An overview is given of a series of recent experimental cases of ESCA application for the characterization of different metals and metallic alloys, illustrating the main results and various phenomena, such as the formation of impurity defects, corrosion, migration of constituent elements in various alloys, clustering in liquid alloy, etc., that can occur on the surface and the interface of investigated materials. These materials comprise the collection coins of noble metals, some metal alloys and Ni-based superalloys, nitride coatings on stainless steel, composite material with TiAlV alloy, treated austenitic steels, and graphene interface with polycrystalline metal foils. The present review could be particularly recommended for the newcomers to the research field of surface analysis and its application for various metals, their treatments, and possible modifications in operating conditions.

**Keywords:** surface analysis; metals and alloys; metal coatings; XPS; AES; SPEM

## 1. Introduction

From the very beginning of surface science around 1960, coinciding with the discovery of electron spectroscopy for chemical analysis (ESCA) [1], a great part of the first surface analysis studies has been dedicated to various metals [2]. This is easily understandable because the metals are stable in ultrahigh vacuum (UHV), and their surface is relatively clean (or can be easily cleaned) and is not modified under soft X-rays or electron beam. Therefore, during the initial boom of surface analysis—namely when the main experimental techniques were developed, the main principles were established, and spectroscopic catalogues were created—great attention of this research was given to the surface of metals, and in particular, the transition metals and noble ones. In this period, new scientific journals dedicated to surface analysis were also born, such as *Journal of Electron Spectroscopy* and *Related Phenomena*, etc. Classical examples of metals' surface studies can be found already in the first volume of the first journal mentioned [3,4]. Even later, when surface analysis became a common tool in the labs of materials characterization and when the ESCA handbooks of all chemical elements were available [5,6], the application for the surface of metals remained an important research field, as it was emphasized in the first text books on surface analysis [7,8].

Currently, when a great variety of surface-sensitive electron spectroscopies (see [9]) are available everywhere in the world, the most used ones remain X-ray photoelectron spectroscopy (XPS) and Auger electron spectroscopy (AES) that were born with a common term of ESCA. Of course, at present time, surface analyses are aimed at more sophisticated materials than elemental metals, but

even for elemental metals these techniques are widely used for the simple and reliable control of surface purity in the fields of materials research and technological applications. It is namely for these reasons that we decided to prepare a short review, illustrating the importance and analytical capabilities of surface analysis for the exploration of metals, including also their modifications induced by operating conditions. In this review, we present the most interesting cases of experimental research carried out in our lab during the last few decades. These cases comprise the surface defects on noble metals (collection coins), some metal alloys and superalloys, nitride coatings on steel, composite material with TiAlV alloy, treatments of austenitic steel, and graphene growth on polycrystalline metals. The common denominator of all these cases is the application of ESCA, i.e., XPS and AES techniques, for materials' characterization.

The main working equations of ESCA are very simple, and they are based on the principle of energy conservation. In the case of XPS, this equation is

$$BE = h\nu - KE - WF, \tag{1}$$

where BE is the binding energy of the elemental core level, $h\nu$ is the photon energy of X-rays, KE is the kinetic energy of the emitted photoelectron, and WF is the work function of the spectrometer.

In Auger effect, the electrons from three different atomic levels are involved, resulting in the finally excited Auger electron with kinetic energy equal to the following:

$$KE = EL1 - EL2 - EL3 - WF, \tag{2}$$

where KE is the kinetic energy of emitted Auger electron; EL1 is the binding energy of the first atomic level, where the hole is created (by X-rays or electron beam); EL2 is the energy of the second level from which the electron is falling down to the lower level EL1; and EL3 is the energy of the third atomic level from which the Auger electron is ejected. Of course, in some chemical elements the last two levels of the Auger process can be located in the valence band, where a core–valence–valence (CVV) Auger peak is then observed. Typical examples of such elements with broad Auger CVV peaks are carbon and silicon. A schematic diagram of the final state in photoemission and Auger excitations is illustrated in Figure 1.

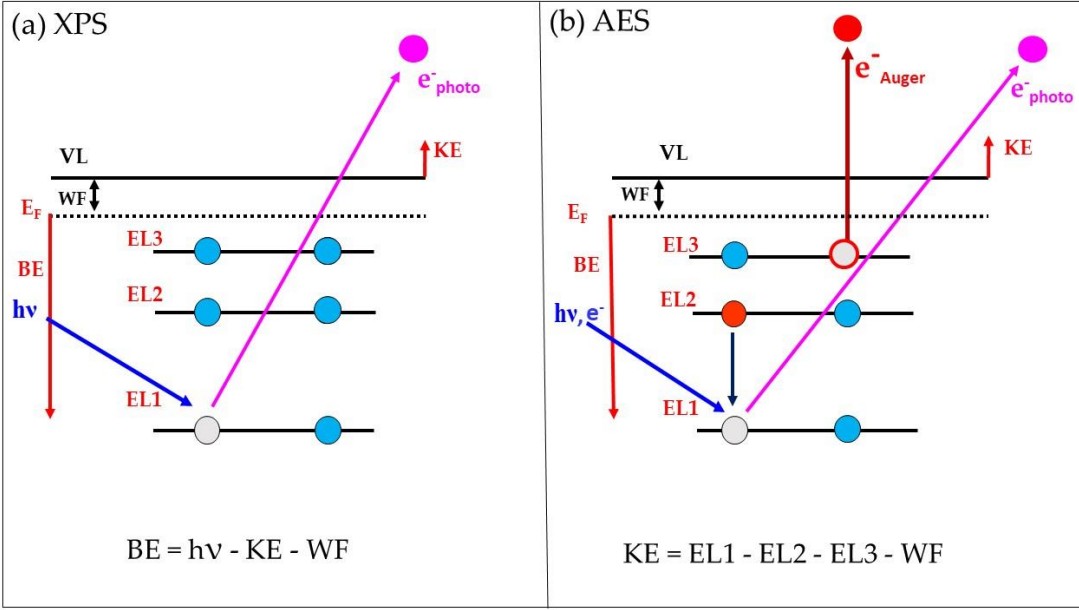

**Figure 1.** Schematic diagram of the final state in photoemission (**a**) and Auger (**b**) excitations.

From the photoemission and/or Auger spectra, it is possible to identify the chemical elements because the energy of these peaks is characteristic for every element. In the case of superposition of some peaks from different elements, other peaks of the same elements can be used for identification. In most cases, the XPS also permits the chemical state of constituent elements to be identified due to the chemical shift of the photoemission peaks [10]. Both the techniques are surface sensitive because their information depth is limited by the mean free path of the electrons in the solid, which depends on kinetic energy and is typically from 1 to about 10 nm. The detailed description of XPS and AES techniques can be found in numerous text books (e.g., [7–9]) and even online (e.g., [11]).

In the case of AES, the primary electron beam can be easily focused as with electron microscopy, and therefore it is possible not only to achieve a high lateral resolution in spectroscopy but also to acquire the chemical maps of the surface. This mode of operating is called Auger scanning microscopy (SAM). The first two generations of XPS spectrometers were equipped with standard soft X-ray sources (typically with Al and Mg anodes); because of this, the focusing of X-rays was impossible, and the lateral resolution of these instruments was limited to about 0.1–1 mm. Later on, with the introduction of monochromatized X-ray sources and electromagnetic input lenses, the lateral resolution of XPS was improved to about 1–3 microns, allowing also for the operation in XPS imaging mode, i.e., to acquire the surface chemical maps [12,13]. However, quite often this resolution can be too low for the investigation of the submicrometric features or patterns on the sample surface. A much higher lateral resolution of photoemission spectroscopy and imaging can be achieved by using the dedicated beamlines of synchrotron radiation. This technique, which is called scanning photoelectron microscopy (SPEM), enables us to investigate the surface chemical composition at a lateral resolution of about 100 nm [14,15]. It was successfully employed also for some of our experimental cases by using an ESCA microscopy beamline at the synchrotron Elettra in Trieste, Italy. The main features of XPS, AES, and SPEM techniques are summarized in the Table 1, including also the benefits and weak points of their practical applications.

**Table 1.** Comparison of the main features of three ESCA techniques.

| Features | XPS | AES | SPEM |
|---|---|---|---|
| Probe | Soft X-rays | Electrons | Soft X-rays |
| Spectroscopic signal | Photoelectrons, Auger electrons | Auger electrons | Photoelectrons, Auger electrons |
| Detectable elements | Li and higher | Li and higher | Li and higher |
| Sampling depth | 0.5–10 nm | 0.5–10 nm | 0.5–10 nm |
| Detection limit | $1 \times 10^{-4}$ | $1 \times 10^{-3}$ | $1 \times 10^{-4}$ |
| Information | Elemental, chemical | Elemental | Elemental, chemical |
| Quantification | OK | semi | OK |
| Lateral resolution | >3 µm | >30 nm | >50 nm |
| Advantages | Chemical bonding, no sample damage | High resolution of chemical imaging | Chemical bonding, chemical imaging |
| Disadvantages | Poor lateral resolution | Sample charging, beam-induced damage | Beam-induced damage |

## 2. Experimental Techniques

Two different spectrometers were used for the XPS characterization of investigated materials: an aged Escalab MkII (VG Scientific Ltd., East Grinstead, UK) and a modern one, Escalab 250Xi (Thermo Fisher Scientific Ltd., East Grinstead, UK). In both the instruments, the spectroscopy was carried out by concentric hemispherical analyzers operating in a constant pass energy (20 or 40 eV) mode. The first one was equipped with a double-anode (Al/Mg Kα) X-ray source and electrostatic input lens,

collecting the signal from the sample area of about 10 mm (large-area mode) and variable to about 0.3 mm (small-area mode). The photoemission signals were registered by a 5-channeltron detector. The second apparatus was equipped with a monochromatized Al K$\alpha$ source and a combined system of electrostatic/electromagnetic input lenses. In the spectroscopy mode, this system allowed the diameter of analyzed sample area to vary from 900 to 20 µm, and the photoemission signals were registered by a 6-channeltron detector. In the imaging XPS mode, the best lateral resolution of chemical maps was of about 3 µm, and the signals were registered by a multichannel plate with 128 channels. The charging of insulating samples was suppressed by using a combination of two neutralizing floods: low energy electrons from an in-lens gun and low energy Ar+ ions from an external gun. For the sample surface cleaning and XPS depth profiling in both the Escalabs, rastered Ar+ ion guns were used, i.e., the EX05 model in MkII and the EX06 in 250Xi. The base UHV pressure in the analysis chambers of both spectrometers was always kept below $10^{-9}$ mbar.

The experiments of AES/SAM were carried out by using a LEG200 electron gun installed on the analysis chamber of Escalab MkII. This excitation source provided the primary beam of electrons with an energy up to 10 keV and a minimum beam diameter of 200 nm. For all the samples, the current of electron beam was kept very low (4–10 nA) in order to avoid any sample surface damage by the electron beam. Seeking to increase the signal-to-noise ratio, all the Auger spectra and chemical maps were acquired in a constant retard ratio (1:2) mode of the analyzer.

All experimental data were processed by the software Avantage v.5 (Thermo Fisher Scientific Ltd.). The peak fitting of photoemission spectra was performed by using the Shirley background, a Voigt peak-shape (mixed Gaussian-Lorentzian with variable ratio), and linked full widths at half maximum (FWHMs) for the same core level. Final calibration of the BE scale was done by fixing the main component of C 1s peak (aliphatic carbon) at 285.0 eV and controlling it, if the Fermi level in the valence band is positioned at BE = 0.0 eV.

High resolution SPEM experiments were performed at the ESCA microscopy beamline of the Elettra synchrotron [14,15]. By using Fresnel zone plate optics, the X-ray beam from the synchrotron source was focused to a microprobe with a diameter of about 150 nm on the sample, which was raster-scanned with respect to the microprobe. Photoelectrons were collected by the SPECS-PHOIBOS 100 hemispherical analyzer and registered by a 48-channel electron detector. All the samples were investigated in both imaging and spectroscopy modes with a 0.2 eV energy resolution by using 500–700 eV photon energy. The overall lateral resolution was below 50 nm. Before the measurements, the samples were cleaned by Ar$^+$ ion sputtering at 2.0 keV energy. After the acquisition, the chemical maps were processed by the Igor v.6.3 software.

## 3. Experimental Cases of Different Materials

### 3.1. "Gold Corrosion" in Collection Coins

Can "gold corrosion" occur in gold coins? This question arose approximately two decades ago, when some owners of precious coins unexpectedly found the appearance of numerous stains on their gold coins. After many studies, even using the Pourbaix diagram, this enigma was successfully disclosed because of the application of surface analysis techniques. The chemical composition of these defects was determined, and their source was established.

The study of surface analysis was performed on gold and silver collection coins supplied by the Kunsthistorisches Museum in Vienna (historical Austrian Ducat) and Austrian Mint (coins and their blanks). XPS, AES, and SAM techniques were combined to get qualitative and quantitative information about the surface defects. Their stains, analysed by a stereomicroscope, were generally composed by a dark central area surrounded by a larger outer area, whose colour varied from red to dark blue [16]. The chemical composition of every single stain was determined by XPS. All elements in the spot were quickly identified by the assignment of the peaks found in the survey scan spectrum

(Figure 2), whereas their chemical state and atomic concentration were determined by processing the resolved spectra of the main peaks presented in Figure 3 [17].

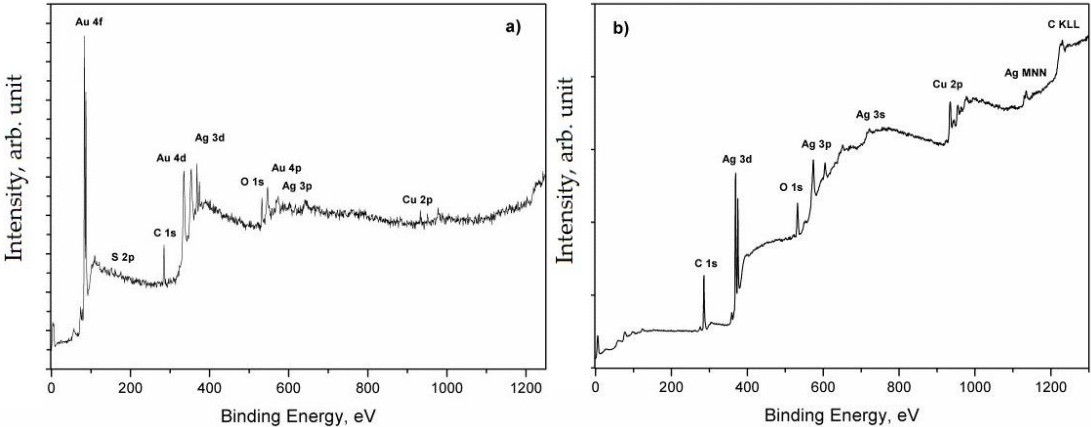

**Figure 2.** XPS survey scans of the gold (**a**) and silver (**b**) coins.

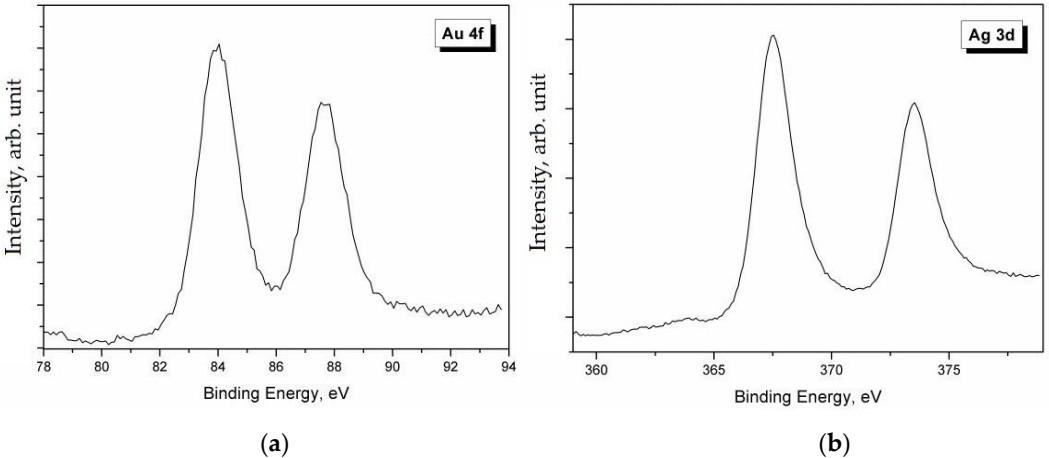

(**a**)                                          (**b**)

**Figure 3.** High-resolution XPS spectra of Au 4f (**a**) and Ag 3d (**b**).

The obtained results promptly evidenced a strange composition of the stains on a pure (999.9) gold coin: A contamination with Ag and S was revealed. This was an astonishing finding for a pure gold coin, giving rise to the following questions: how and when had these impurities been added? The obtained results were confirmed by the multipoint AES analysis and SAM chemical maps acquired with a higher lateral resolution of approximately 200 nm, which are presented in Figure 4.

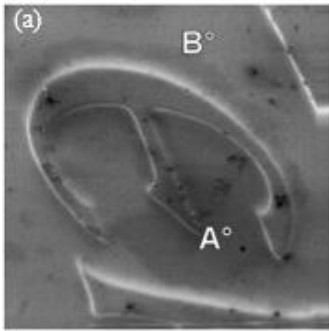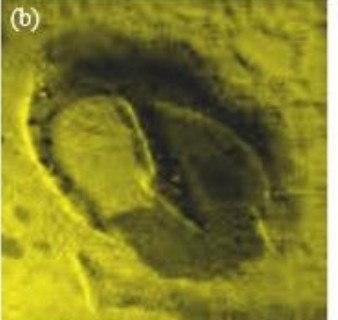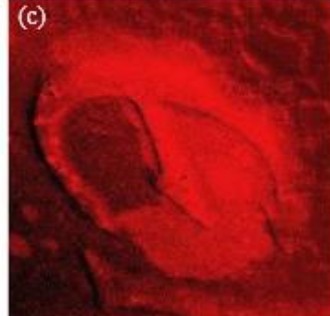

**Figure 4.** *Cont.*

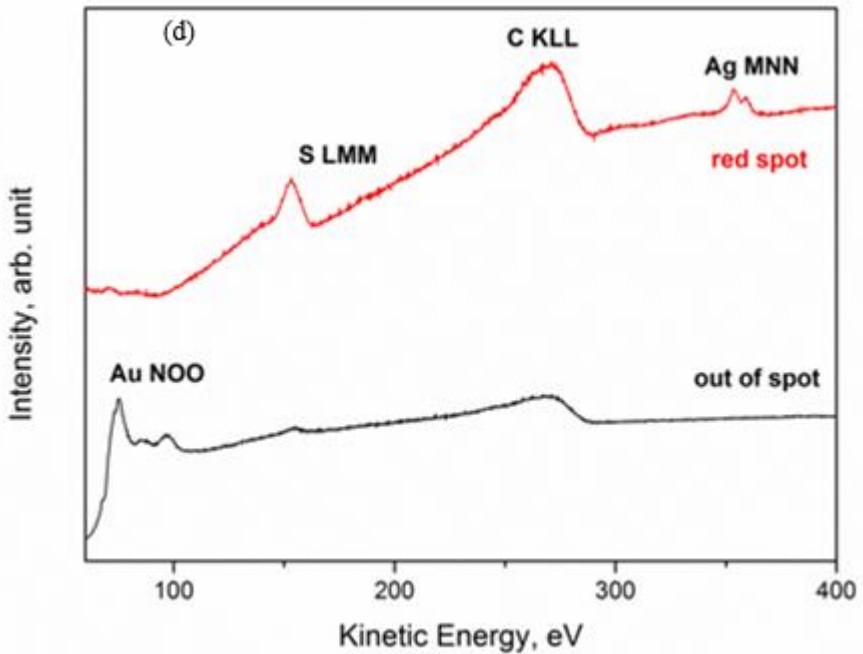

**Figure 4.** Auger electron spectroscopy (AES) investigation of the defect on gold coin: SEM image—size $1.8 \times 1.8$ mm$^2$ (**a**); Au NOO image (**b**); Ag MNN image (**c**) and AES spectra (**d**) acquired in the red spot (point A) and out of the spot (point B) [17].

The analysis of Ag 3d, Ag LMM, and S 2p spectra gave some indications on their chemical state. As it can be seen in Figure 4, the Ag 3d spectrum was characterized by the typical doublet of the spin-orbit splitting of the core level 3d (Ag 3d$_{5/2}$–Ag 3d$_{3/2}$), separated by 6.0 eV. The main Ag 3d$_{5/2}$ peak was positioned at BE = 368.0 eV. However, it is well known that the Ag 3d signal is one of the few cases where the chemical shift is almost absent, i.e., it is impossible to identify the chemical state of Ag only from photoemission spectra. In these cases, it is necessary to calculate the modified Auger parameter $\alpha'$ by using a very simple formula: $\alpha'$ = BE (Ag 3d$_{5/2}$) + KE (Ag LMM) [18]. The value of the Auger parameter can indicate the chemical state (metal, oxide, etc.) of the investigated element. In this case, it was $\alpha'$ = 725.2 – 725.3 eV, which is the typical value for Ag$^+$ in the silver sulfides, specifically in Ag$_2$S [5]. The analysis of the S 2p signal confirmed the presence of sulfides, since the S 2p$_{3/2}$ peak was positioned at BE = 161.6–161.9 eV [5].

It is interesting to note that the XPS quantitative analysis identified four different scenarios, depending on the color of the spot, which are summarized in Figure 5: (1) grey stains—with Ag, O and S; (2) dark spots—with Ag, O, and S, but also with Au and Cu; (3) red spots—like the grey spots, but with different atomic concentration of the elements; and (4) clean surface—with Au, Cu, and a small amount of O.

Then, the different chemical composition of the stains was investigated by XPS depth profiling, which revealed the different thicknesses of the stains: from 5 to 6 nm for red ones to about 300 nm for dark blackish colored ones. Therefore, the variation of the color was principally related to a different thickness of contamination layer, where the thickness of Ag$_2$S was always limited to the first 3–5 nm and the second sublayer of metallic Ag continued in depth. These results suggest that a thin, almost transparent, overlayer of sulphide was formed by the interaction of metallic Ag with the sulfur-containing contaminants in air (like H$_2$S), whereas some bigger silver particles were mechanically embedded into the coin surface during the milling, rolling, or punching of the gold strips.

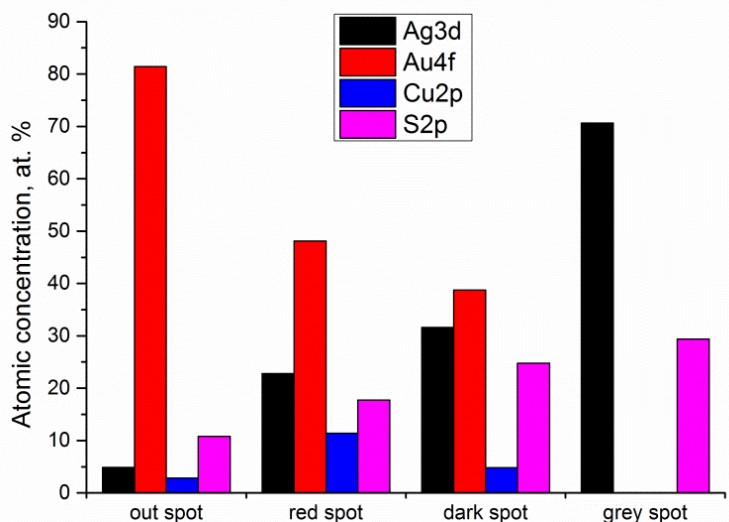

**Figure 5.** XPS elemental quantification of the different spots on the gold coin.

### 3.2. Hard Coatings of Nitrides

Hard coatings, based on transition metal nitrides or carbides, are characterized by excellent mechanical properties, suited for steel protection and fabrication of cutting tools. Their performance is continuously improved by the optimization of the fabrication processes, the development of new deposition technologies, and the production of composite materials with enhanced physical and chemical properties. An important contribution to the development of these coatings can be given by the use of surface analysis, which enables us to find the best production conditions and to improve their quality. In this section, the results obtained by the XPS and AES investigations of the TiN-Ti composite and a multilayer CrN–Cr coating are presented. As it can be seen in Figure 6, the deconvolution of the Ti 2p spectrum shows the presence of multiple contributions due to the different chemical states of Ti: the components 3 and 4 located at BE = 458.5 and 456.5 eV were assigned to the chemical states of $Ti^{4+}$ and $Ti^{3+}$ bound to oxygen; the component 1 positioned at BE = 454.1 eV was related to metallic Ti (0); finally, the component 2 positioned at BE = 455.0 eV was assigned to the bonds of Ti–N and Ti–C [19,20].

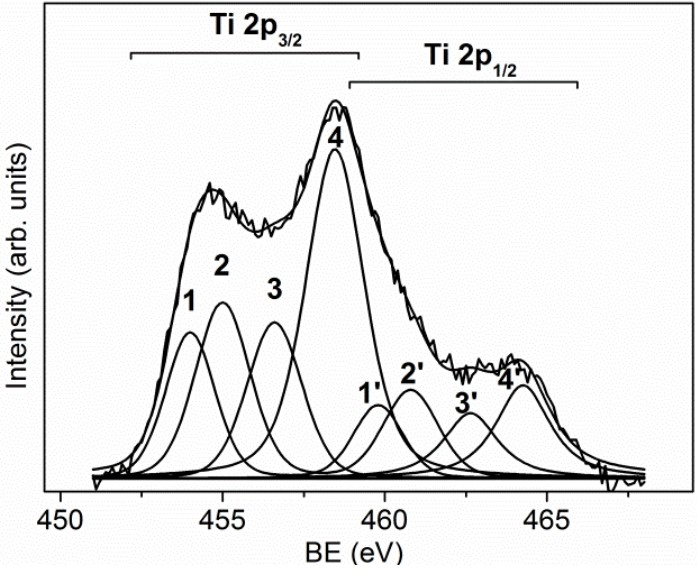

**Figure 6.** Ti 2p peak fitting of the Ti/TiN composite coating [19].

Naturally, the presence of oxides was caused by the oxidation of metallic Ti in air. After ion sputtering, they were almost removed as it is shown in Figure 7. Of course, the possible influence of the preferential sputtering of oxygen [21,22] to the reduction of oxides cannot be excluded, but in our case this effect was not considered, as this study aimed to determine the composition in the volume of Ti nitride after removal of the native oxides overlayer. By using XPS depth profiling, i.e., alternating cycles of ion sputtering and spectra acquisition, it is possible to investigate the changes of chemical composition until a depth of about 1 μm. From the depth profile shown in Figure 8, it is possible to observe how the content of oxides decreases in depth, whereas the contents of metallic Ti and nitrides increase. This trend was also confirmed by the depth profile of the N 1s signal, which was composed of two peaks positioned at BE = 397.0 and 399.5 eV and was assigned to the bonds of N–Ti and N–O in oxynitride compounds, most probably formed due to environmental contamination. The atomic ratio Ti/N = 3.5 was constant along the depth profiling. This excess of Ti content indicated the formation of composite TiN-Ti.

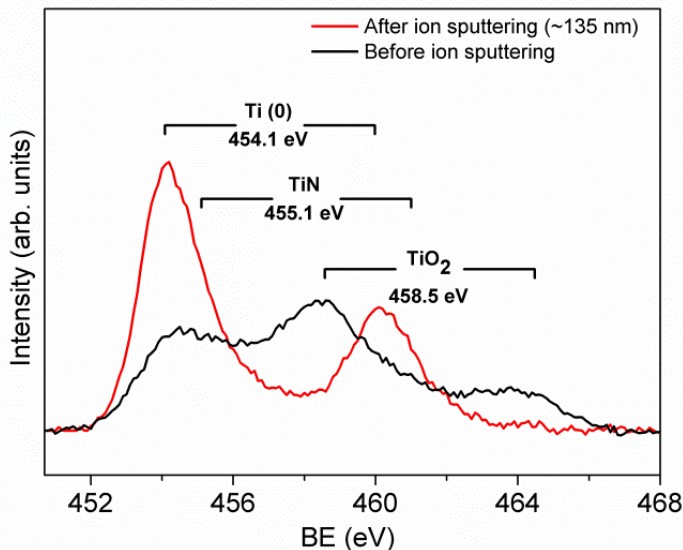

**Figure 7.** Comparison of the Ti2p signals acquired before and after Ar$^+$ ion sputtering [19].

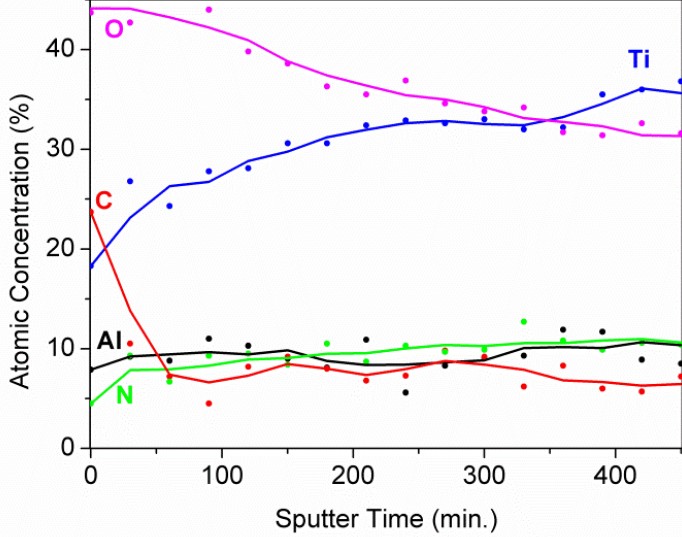

**Figure 8.** XPS depth profile of the Ti/TiN composite coating. The average sputtering rate was equal to 0.3 nm·min$^{-1}$ [19].

Due to the limited depth of XPS depth profiling, the study of the multilayer coating CrN/Cr/CrN with the thicknesses of 1.5/1.0/1.5μm was carried out only for the top layer of this coating [23]. In this layer, the signal of Cr 2p (Figure 9) was composed of a typical Cr $2p_{3/2}$–$2p_{1/2}$ doublet, which was positioned at BE = 574.2 eV, and a large peak due to the contribution of multiplet splitting, centered at BE = 576.0 eV. The deconvolution of the N 1s spectrum evidenced the presence of two chemical species: chromium nitride at BE = 397.1 eV and a component of oxynitrides at BE = 398.6 eV, probably due to the presence of a low amount of oxygen in the deposition chamber. The obtained BE values of N 1s and Cr $2p_{3/2}$ (single component) indicated the formation of CrN, excluding the phase of $Cr_2N$ characterized by a noticeably higher value of BE [5]. This supposition was confirmed also by the determined atomic ratio of Cr/N nearly at 1.0. The XPS depth profile, depicted in Figure 10, showed that after removal of the surface contamination, the composition of CrN coating remained almost constant. Since the total thickness of the coating (~4 μm) was too high for XPS depth profiling until the substrate, it was stopped after the removal of ~100 nm of CrN, and the cross section of the coating was further investigated. Due to the limited lateral resolution of XPS, the interfaces of CrN/Cr, Cr/CrN, and CrN/substrate were investigated by the AES/SAM technique.

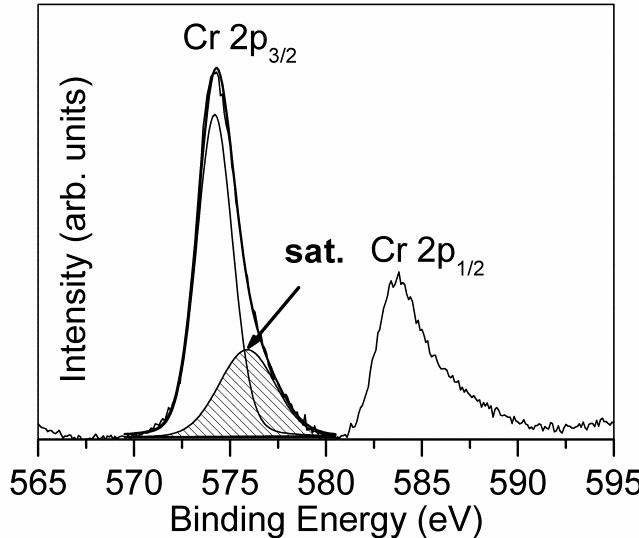

**Figure 9.** Peak fitting of Cr 2p spectrum acquired on the top of the multilayer coating [23].

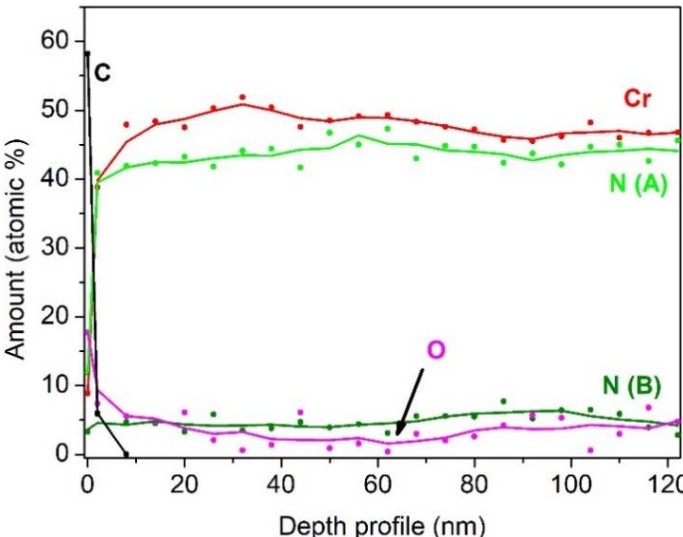

**Figure 10.** XPS depth profile of the first CrN layer in composite coating. The average sputtering rate was equal to 0.3 nm·min$^{-1}$ [23].

Figure 11 shows the SEM image and the multipoint AES analyses carried out on the cross section of the sample. The AES spectra were acquired on different points, moving from the substrate (region 1) to the top of the coating (region 4). The substrate was characterized by the presence of Fe LMM peaks (KE = 594.0, 652.0 and 705.7 eV) and the low-intensity KLL peaks of C and O (see Figure 11b). In Regions 2 and 4, the peaks of Cr $L_3M_{23}M_{45}$ (KE = 530.6 eV) and N KLL (KE = 385.4 eV) were registered, whereas in Region 3, only a peak of Cr $L_3M_{23}M_{45}$ was present. In addition, the chemical maps were acquired by SAM, where the investigated area of the sample was represented by pixels of the peak-minus-background intensity of the selected Auger peak. The SAM images collected by using the peak-minus-background of the Cr $L_3M_{23}M_{45}$ and N KLL peaks are shown in Figure 12. The black points indicate the area without signal, whereas the lighter grayscale points indicate the area where the signals were detected. As it can be noticed, the layers are well-defined, and the interface is rather neat, suggesting the absence of diffusion phenomena during the deposition process. The coating thickness, estimated from the SEM/SAM images, was about 4.0 μm.

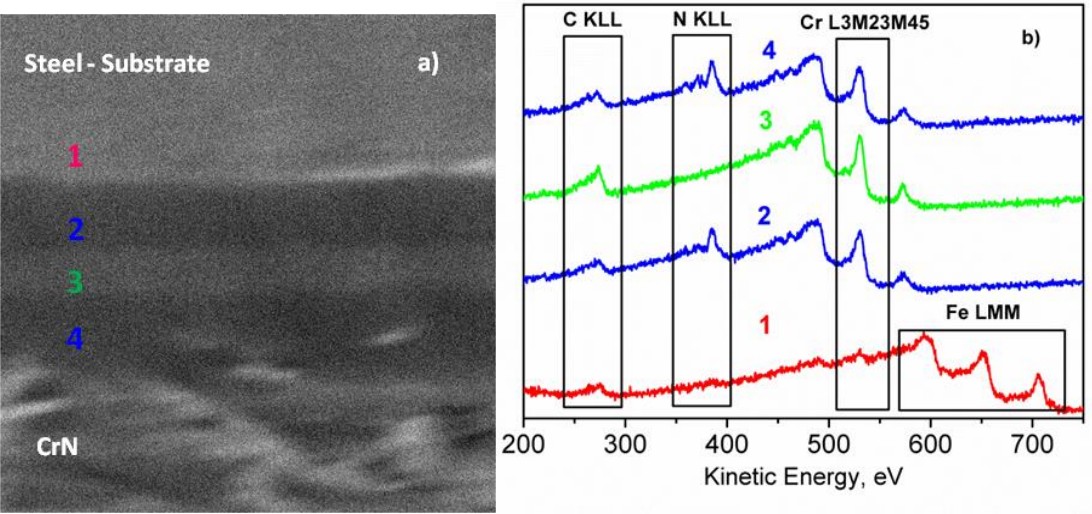

**Figure 11.** SEM image (**a**) and AES spectra (**b**) acquired along the cross section of the multilayer coating [23].

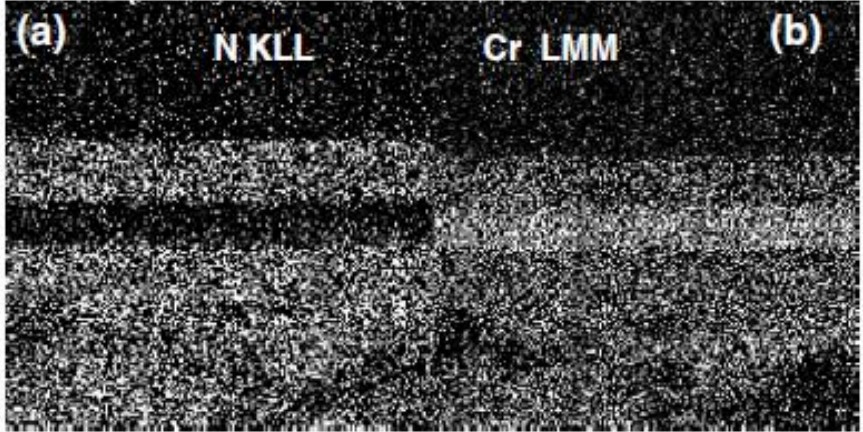

**Figure 12.** N (**a**) and Cr (**b**) chemical maps of the cross section of multilayer coating [23].

*3.3. Microchemical Composition of Ni-Based Superalloys*

　　　Superalloys are a class of materials that find numerous applications in the metallurgical field, in particular when a high strength, superior oxidation, and corrosion resistance at temperatures above 700 °C are aquired. Many of superalloys properties are determined by their microstructure, and therefore it is quite important to predict the microstructural evolution during long-time operation, especially the coarsening and morphological changes of the γ′ phase that take place at the operating temperature of 800–900 °C. These superalloys are composed of cuboidal γ′ particles with submicrometric dimensions, embedded in the γ matrix. The chemical composition of two phases could be different, but most of the previous experimental studies have been dedicated to the morphology and microstructure of superalloys, e.g., [24,25] and the references therein. Practically, the data on the chemical composition of the two phases in various superalloys are absent in the literature. However, the coarsening of γ′ particles strongly depends on the difference of chemical composition between a disordered matrix and cuboidal particles. Since this change must occur at the microscale, the surface investigations of the microchemical structure of a biphasic (γ + γ′) Ni-based CM186 superalloy were performed at a high lateral resolution by using the laboratory of scanning photoemission microscopy (SPEM) at the Elettra synchrotron (Trieste, Italy). This technique allows us to directly acquire the surface chemical maps of constituent elements and to determine the variation of their atomic concentrations, eventually induced by the creep tests. In order to prepare for SPEM investigations, the XPS spectra were collected and processed by using a standard XPS apparatus [26,27].

　　　The spectral region, containing all 4f photoemission peaks of constituent elements together with the overlapping peaks of W 5p and Re 5p, is shown in Figure 13.

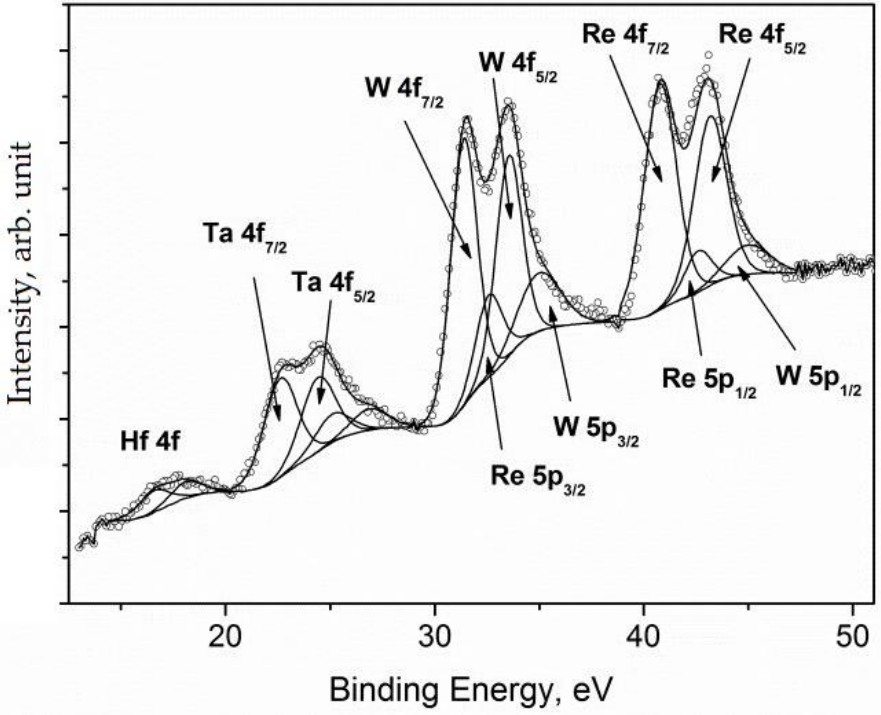

**Figure 13.** XPS spectrum of the 4f region acquired for the sample of CM186 superalloy [26].

　　　The peak fitting analysis revealed that Re $4f_{7/2}$ and W $4f_{7/2}$ peaks were located at BE = 40.8 and 31.4 eV, corresponding to their metallic states, whereas the Ta $4f_{7/2}$ peak was characterized by two components at 22.6 eV and 25.1 eV, assigned to metallic and oxidized species [26], respectively. Finally, the peak of Hf $4f_{7/2}$ at BE = 16.5 eV was assigned to oxidized species [26].

The chemical maps were recorded in different zones of the samples before and after the creep test, shedding light on the compositional differences between γ and γ' phases. After the acquisition, each map was numerically processed in order to remove the contribution of surface morphology from the photoemission signals. It is worth noting that in SPEM the chemical images can be acquired without any chemical etching of the samples, which is the contrary of the standard microscopies (SEM, AFM, etc.) used for superalloys. Figure 14 shows some examples of obtained chemical maps. The Re 4f, W 4f, and Ta 4f images were acquired from the interdendritic zone on the as-received sample.

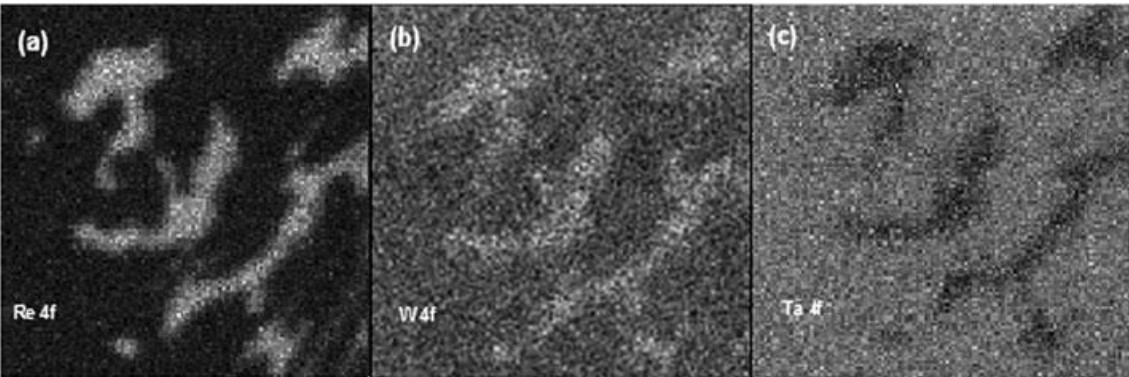

**Figure 14.** Re 4f (**a**), W 4f (**b**), and Ta 4f (**c**) chemical maps (6.4 × 6.4 μm²) acquired on the interdendritic zone of the as-received sample [26].

The chemical maps of Re and Ta were complementary, namely, the bright zones in the Re map corresponds to the black zones in that of Ta and vice versa, whereas the tungsten was distributed homogeneously through the analyzed area, even if its content was slightly higher in the γ phase. The lateral distribution of Re and Ta did not change in the crept sample (Figure 15), since they were concentrated in γ and γ' phases, respectively. In comparison with the as-received sample, the distribution of W after creep appeared more uniform. The relative distribution of the main constituent elements between the γ and γ' phases in the as-received and crept samples is displayed in Figure 16. Each data point is the average value of 5 measurements carried out on different points of the same phase. As it can be noticed, both the phases were characterized by the same amount of Ni, while the concentration of Co and Re was predominant in the γ phase.

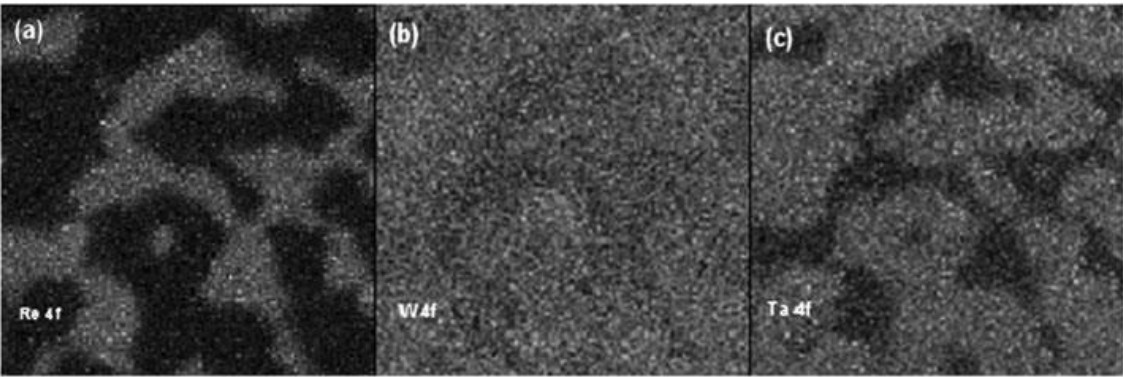

**Figure 15.** Re 4f (**a**), W 4f (**b**), and Ta 4f (**c**) chemical maps (5.1 × 5.1 μm²) of the sample after the creep test [26].

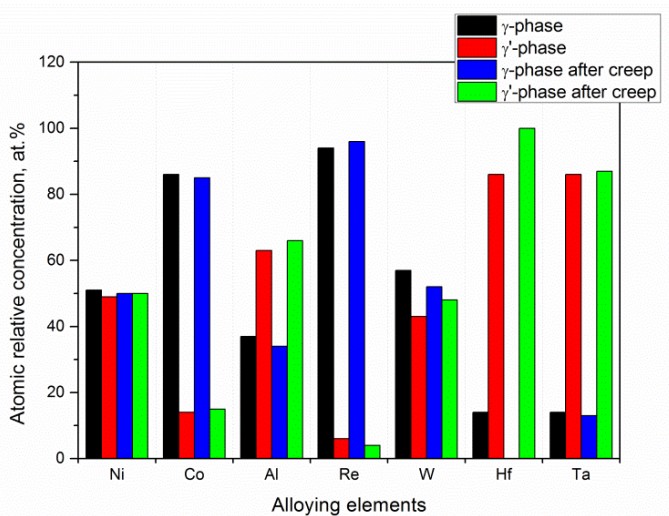

**Figure 16.** Elemental distribution between the γ and γ′ phases before and after the creep test.

After the creep test, their distribution remained almost the same. On the contrary, the amount of Al and Ta was predominant in the γ′ phase, remaining unchanged after the creep test. Significant differences were found for W and Hf, where the creep test induced a migration of these elements from the γ phase to γ′ phase. The obtained results evidenced that this diffusion process is responsible for the weakening of the disordered matrix during the creep.

### 3.4. Diffusion Phenomena in the $Ti_6Al_4V/SiCf$ Composite

There are only a few analytical techniques capable of investigating the diffusion mechanism of the elements in a solid-state sample. Among them, the surface analysis techniques represent the most powerful tool of the investigation, especially in the proximity of the interface between different materials. In this section, we illustrate the multitechnique approach applied for the investigation of a composite material consisting of a $Ti_6Al_4V$ matrix and SiC fibers [28–32].

To avoid the formation of brittle compounds like $Ti_5Si_3$ at the interface matrix/fiber, each fiber was coated with a 3 μm thick graphite layer. However, at the high temperatures during the fabrication process and in-service life, some elemental diffusion could be induced, reducing the mechanical performance of the composite. Figure 17 shows the elemental distribution on the cross section of the sample.

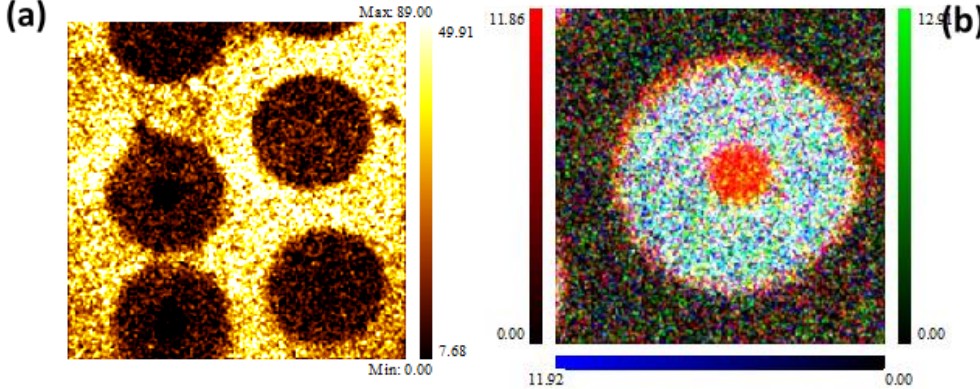

**Figure 17.** XPS images: (**a**) Ti 2p—400 × 400 μm$^2$; (**b**) Si 2p (blue), C 1s—190 × 190 μm$^2$, carbide (green) and graphite (red) [29].

The XPS chemical maps were acquired by collecting the intensity of the signals positioned at BE = 458.8 eV (Ti 2p$_{3/2}$), BE = 529.0 eV (O 1s), BE = 99.9 eV (Si 2p), and the intensity of C 1s, where the contributions of graphite (BE = 284.6 eV) and carbide (BE = 283.0 eV) were separated. As it can be seen, the fibers were embedded in the Ti$_6$Al$_4$V matrix, which the surface contained oxidized Ti species due to the reaction with atmospheric oxygen. The layer of titanium oxides was promptly removed after a brief time of ion sputtering, reducing the Ti chemical state to metallic one. Unfortunately, the lateral resolution of the standard XPS imaging (>3 μm) was too low for us to investigate the diffusion processes that can occur at the interface matrix/fiber. Therefore, the investigation at a higher lateral resolution was performed by an AES multipoint analysis. SEM images and AES line scan spectra acquired for Samples 1 (as prepared) and 2 (heated for 1000 h at 600 °C) are displayed in Figures 18 and 19, respectively.

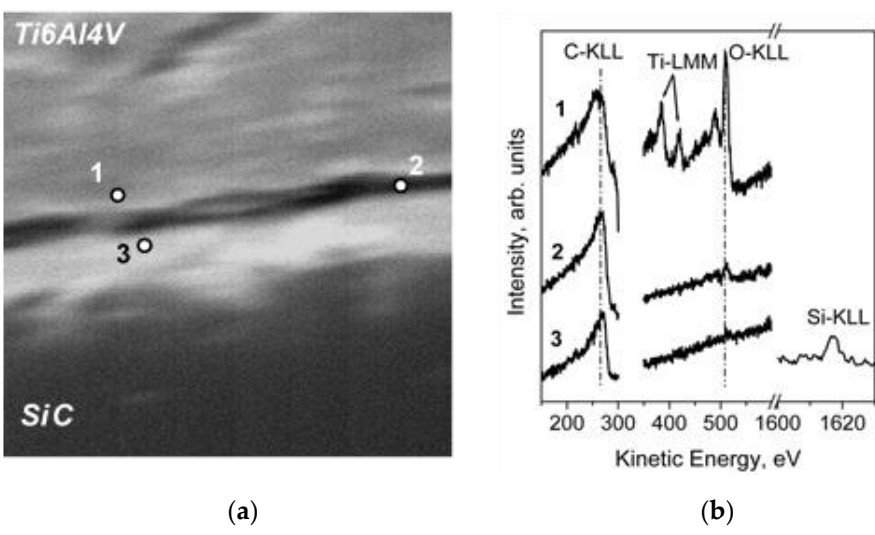

(**a**)                                        (**b**)

**Figure 18.** SEM image 80 × 80 μm$^2$ (**a**) and AES spectra (**b**) acquired on the cross section of the sample 1 across the carbon layer; analysis points are labelled 1, 2, and 3 [29].

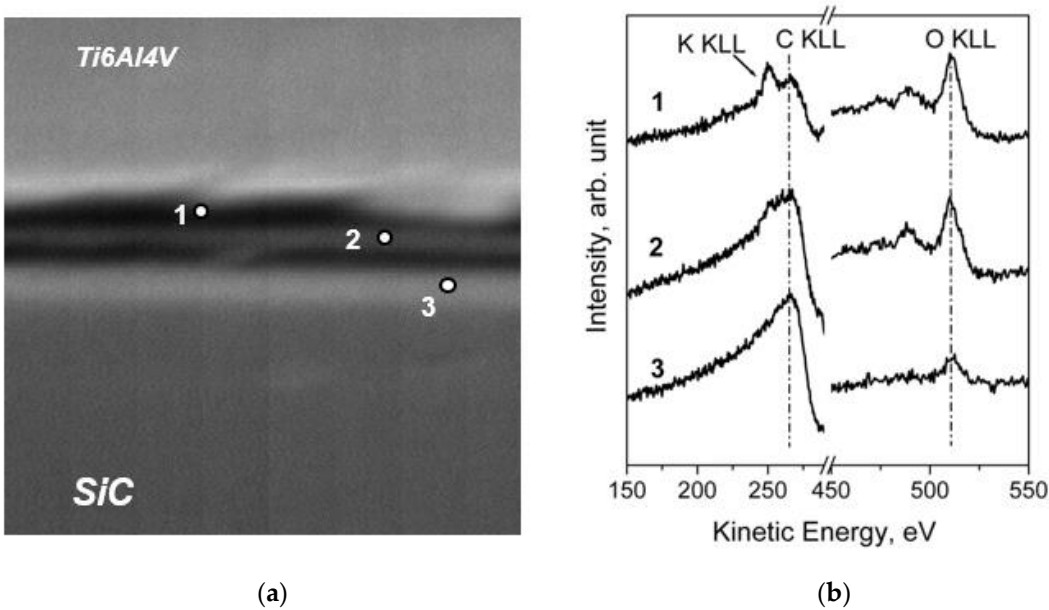

(**a**)                                        (**b**)

**Figure 19.** SEM image 80 × 80 μm$^2$ (**a**) and AES spectra (**b**) acquired on the cross section of Sample 2 across the carbon layer; analysis points are labelled 1, 2 and 3 [29].

The obtained results revealed that the graphite layer acts as a good protection barrier, avoiding the diffusion of Si in the Ti matrix. However, as evidenced by the SEM analysis, the morphology of the graphite layer became irregular after a thermal treatment at 600 °C for 1000 h, despite its thickness remaining unchanged. This result can be explained by taking into consideration the reaction between carbon and atmospheric oxygen in producing CO. However, the carbon diffusion in the Ti matrix should also be considered. Since the samples have a curvy geometry, the resolution of standard XPS and AES was not sufficient to characterize the chemical species at the interface. To solve this problem, the interface between the graphite and the metallic alloy was investigated by covering the $Ti_6Al_4V$ and Ti 99.99+ foils with a thin layer of graphite. In order to simulate the diffusion of carbon, the samples were heated in vacuum for 8 h at 500 °C (Figure 20). The XPS depth profiles demonstrated the diffusion of elemental carbon in the metallic matrix, forming a thin layer (about 10 nm) of carbides (Figure 21).

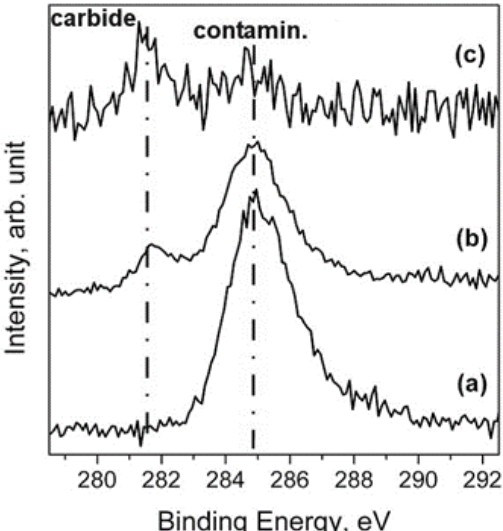

**Figure 20.** Comparison of C 1s spectra of the (**a**) as-prepared Ti 99.99+ sample; (**b**) Ti 99.99+ sample after 30 min of thermal process at 500 °C; (**c**) Ti 99.99+ sample after 140 min of ion sputtering at 1 keV [29].

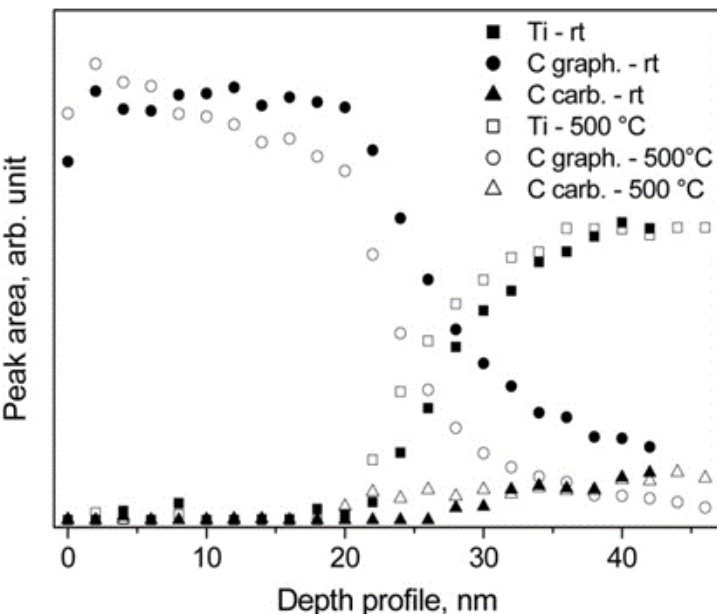

**Figure 21.** XPS depth profiles of the Ti 99.99+ sample covered with carbon at room temperature (rt) and after heating at 500 °C [29].

From the SPEM analyses [32], carried out on the composite samples, it was concluded that the formation of carbides included not only TiC, but also the interstitial–substitutional (i-s) pairs of C–Al and C–V, present in the α phase of the matrix near the fibers.

### 3.5. Microchemical Structure of the PbBi Liquid Alloy

The development of a new generation of nuclear reactors has involved many aspects of material science. One of them was the investigation of the microchemical inhomogeneities occurring at high temperature in a liquid Pb-Bi eutectic (LBE) alloy. LBE finds its application in the nuclear reactor as a coolant and spallation source of MYRRHA, an accelerator-driven system. Therefore, it is quite important to investigate any changes of the microchemical structure of LBE that may induce corrosion and embrittlement phenomena in the structural materials. The microstructure of the LBE alloy was evaluated using high-temperature X-ray diffraction (HT-XRD) [33], whereas the microchemical composition was investigated by SPEM [34–36].

In this section, we focus our attention on the surface analysis. Generally, by SPEM, only the solid samples can be analyzed; thus, in order to simulate the clustering formation, we used a rapid cooling (quenching) of liquid alloy starting from different temperatures and assumed that the microchemical composition of the liquid was preserved on the surface of the obtained solid LBE alloy. The selected temperatures for quenching were 126 °C (eutectic temperature) and 200, 300, 400, 518, and 700 °C. The surface chemical maps were acquired by measuring the intensity of the Pb $4f_{7/2}$ and Bi $4f_{7/2}$ peaks, positioned at BE = 137.0 and 156.0 eV, respectively.

Before collecting the maps, the carbon and oxygen contaminations were removed, operating a short cycle of Ar+ ion sputtering. Although the Pb and Bi native oxides were not completely removed, they were neglected because they are meaningless for this discussion. For convenience, the chemical maps were displayed, indicating the Pb/Bi atomic ratio (AR), which is more representative to the elemental distribution. Taking a reference value of the nominal atomic ratio of the eutectic alloy Pb/Bi = 0.8, three pixel colors were used to evidence the three different cases: (i) blue—lack of Pb with AR < 0.6, (ii) red—excess of Pb with AR > 1.0, and (iii) yellow—near a nominal ratio of 0.6 < AR < 1.0. After the acquisition, each image was processed by applying the following procedure of the Igor software: (1) elimination of morphology effects from Pb and Bi maps by using the correction (peak minus background)/background and (2) the superposition of obtained maps and conversion to the maps of atomic ratio AR. Obtained maps of the AR ($100 \times 100$ μm$^2$ or $50 \times 50$ μm$^2$) processed by MATLAB software are presented in Figure 22. As it can be noted, a strong inhomogeneity was observed. Depending on the quenching temperature, Pb and Bi atoms formed the clusters of different dimensions enriched in Bi and/or Pb. At an eutectic temperature, the surface of the sample was characterized by the presence of micrometer clusters enriched in Bi (~90% of Bi), immersed in the alloy with eutectic composition. Increasing the quenching temperature, the elemental distribution and atomic concentration in the clusters were changed. The clusters size was reduced to a few microns (1–5 μm) as the consequence of higher thermal agitation and these clusters were alternatively enriched in Pb and Bi, and the surface distribution of the alloy with an eutectic composition 0.6 < AR < 1.0 was also changed. The cross-section mapping of the sample that was quenched at 518 °C (see Figure 22f) demonstrates how the cooling process was freezing the sample surface in a structure quite similar to the liquid alloy, while the interior of the sample experienced a different temperature gradient, giving rise to the big clusters enriched in Bi. In order to quantify and compare the elemental distribution in different samples, a statistical calculation of the cumulative area CA was applied:

$$CA(AR_i) = \frac{100}{n} \sum\nolimits_n p_i(AR_i), \tag{3}$$

where n is the total number of selected pixels $p_i$ that have $AR_i$ in the chemical map.

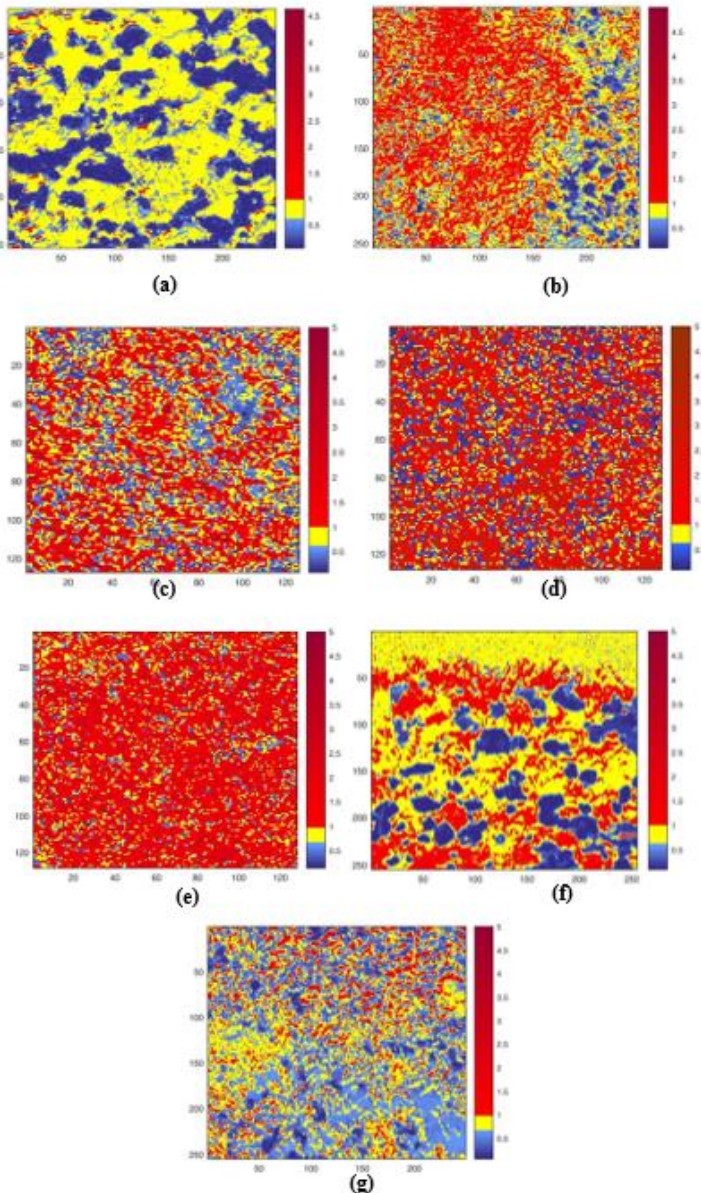

**Figure 22.** Scanning photoelectron microscopy (SPEM) images of the Pb/Bi atomic ratio (AR) distribution for the samples quenched at (**a**) 126 °C, (**b**) 200 °C, (**c**) 313 °C, (**d**) 401 °C, (**e**) 518 °C, (**f**) 518 °C (cross section of the sample), (**g**) 700 °C.

Figure 23 shows the plot of cumulative area (CA) versus the quenching temperature (QT), where the curves were calculated for AR1, AR2, and AR3. At the melting temperature (126 °C), the CA value of AR2 was approximately 2.5%, indicating a very low concentration of Pb, whereas the CA values of AR1 and AR3 were almost similar at 52% and 45%, respectively. Increasing QT, the CA of AR2 was augmenting almost linearly until over 80% at QT = 600 °C, then suddenly falling down below 10% at QT = 700 °C. The comparison of these curves with the phase transition determined by HT-XRD investigations confirmed that the structural modification is also accompanied by the change of the number of clusters enriched in Pb (AR2). As regards the curves of AR1 and AR3, they substantially showed a complementary trend with respect to the AR2 one.

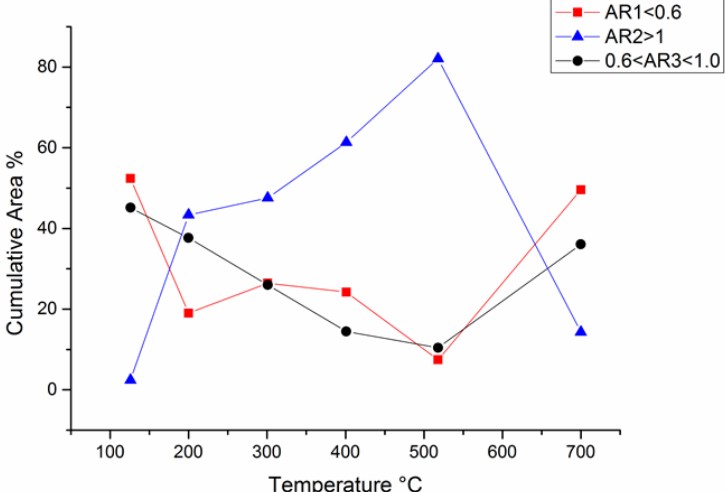

**Figure 23.** Cumulative area (CA) vs. quenching temperature for the three ranges of atomic ratio (AR): the values of AR1, AR2, and AR3.

### 3.6. Austenitic Steels

Austenitic stainless steels are known as materials with a high corrosion resistance in different environments. Because of their low hardness, however, they cannot be used in several industrial applications, unless after modifications through thermochemical surface treatments. Carburizing, nitriding, and carbo-nitriding are common examples of the heat treatments that are used to increase the hardness of stainless steels. These processes need to reach a temperature higher than 550 °C, which could cause the local microstructural changes in the austenitic steel phase, such as the precipitation of Cr carbides. Since these precipitates can reduce the corrosion resistance of the steel, it is necessary to adopt some heat treatment at a lower temperature. A good alternative is the kolstering process, which can harden the austenitic steels without compromising their resistance to corrosion. Although kolstering is a good low-temperature treatment, it is unfortunately very long lasting and expensive. It involves a pretreatment of the steel in an HCl atmosphere at about 250 °C to remove the $Cr_2O_3$ layer from the surface. Then, the stainless steel is treated at 450 °C in a gaseous atmosphere of CO, $H_2$, and $N_2$ for a duration about 30 h.

Very promising results close to those of kolstering were obtained through a plasma carburizing process at low temperature. In the study presented in [37], the plasma was generated by microwaves operating up to 200 mbar as described in detail in [38], while the temperature and pressure were set to about 420 °C and 80 mbar, respectively, for the whole treatment duration of about 6 h. The chamber gas mixture was formed by $CH_4$ (variable percentage) in $H_2$. The main advantage of this treatment is the reduction of the process time, and consequently this is more convenient also for process costs.

XPS and AES techniques allow for the study of the steel surface before and after these treatments. In particular, these techniques permit us to examine the chemical composition of the superficial hardened layer. In this way, it is possible to identify the best process condition, for example, by changing some parameters of the treatment. In this study, the percentage of $CH_4$, added to $H_2$ in the gas mixture was varied from 2% to 10%.

The results of microhardness tests and XRD measurements [1] have established that the sample treated with 2% of $CH_4$ was the one with the best results in terms of hardness (700 HV) and corrosion resistance, without the presence of any precipitates of Cr carbides. For a better understanding of these results, all the samples were investigated by surface analysis.

An AES line scan over the cross section, shown in Figure 24a, revealed the presence of an additional carbon layer with a thickness of about 2–3 μm (lighter zone) above the hardened layer of 20 μm. As it is possible to see from Figure 24b, the Auger signals of C KLL, O KLL, Cr LMM, and Fe LMM were detected along the whole cross section. In the first point, corresponding to the zone near the surface,

the amount of carbon is the highest, and the concentrations of Fe and Cr are low. The ratio of the signals intensity (Fe LMM)/(C KLL) is equal to 0.5. Instead, at the point closest to the bulk, the amount of carbon returns to the nominal value of the alloy, and the ratio Fe/C is equal to 0.9. The intensity ratio of (Fe LMM)/(C KLL) for the entire line scan is shown in Figure 25.

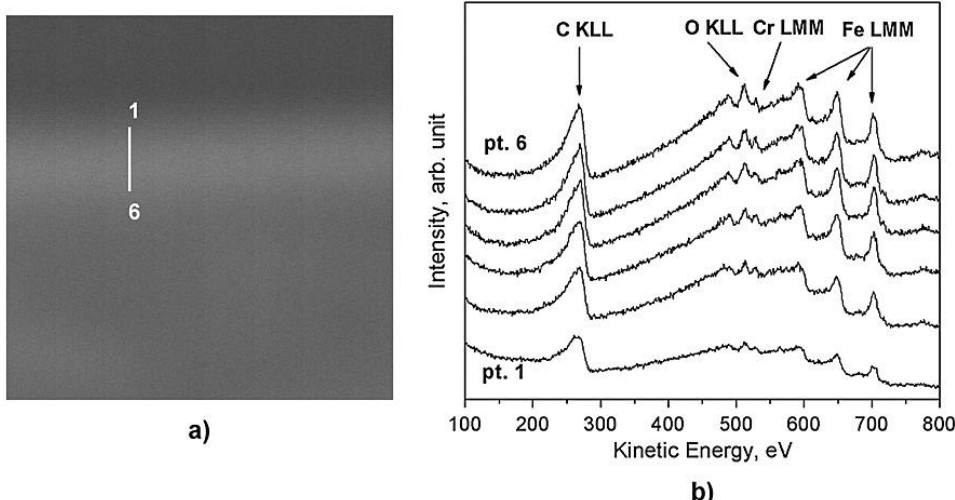

**Figure 24.** SEM image ($24 \times 24$ μm$^2$) of the sample treated by plasma at CH$_4$ 2% in H$_2$ (**a**), and Auger spectra (**b**) recorded in points 1–6 are marked in the SEM image [37].

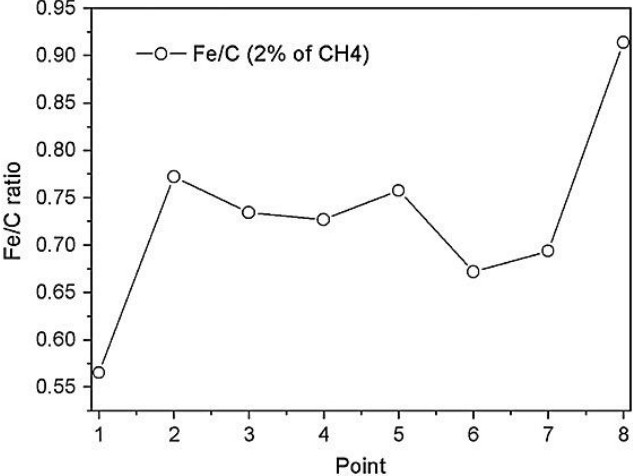

**Figure 25.** (Fe LMM)/(C KLL) intensity ratio calculated along the line scan. [37].

From the value of the D parameter, which is the distance between the most positive maximum and the most negative minimum of the first derivative of C KLL spectrum [39], it is possible to establish that the samples subjected to carburization with a gas mixture composed of 2% of CH$_4$ and H$_2$ have a ultrathin outer layer of graphitic nature (C–C bond with a majority of planar sp$^2$ hybridization). Because of the presence of this additional hard graphitic layer, which was not present in the other samples treated with higher percentages of CH$_4$, it is possible to conclude that 2% of CH$_4$ is the best gas mixture process condition. In this way, the hardened surface of the austenitic steel is comparable to the one obtained with the kolstering treatment.

Another interesting discovery of austenitic steels that has been reported in the papers [40,41], concerns the microstructural modification in the steel with a high content of N (about 0.8 wt.%) induced by heating. Although nitrogen stabilizes the austenitic phase and increases the corrosion resistance, it is important to note that N is soluble only for quantities less than 0.4 wt.% (both in the liquid and solid phase). After exceeding this value, the discontinuous precipitations of chromium nitride are formed in the steel in the range of temperature between 700 and 900 °C.

The transformation that occurs during heat treatments in that temperature range is the following:

$$\gamma_s \rightarrow \gamma + Cr_2N, \tag{4}$$

where $\gamma_s$ is the N-supersaturated austenitic phase (initial phase of the steel), $\gamma$ is the austenitic transformed phase which appears as a lamellar structure, and $Cr_2N$ is the chromium nitride precipitates. A SEM image with the corresponding schematic structure of the austenitic steel is shown in Figure 26.

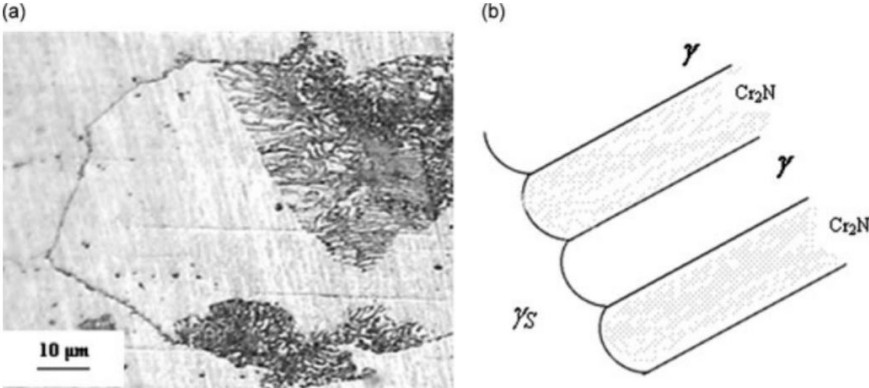

**Figure 26.** SEM image of partially transformed austenitic grain after 2 h at 850 °C (**a**) and a schematic of the structure (**b**) [40].

As it is explained in much detail in the cited papers [40,41], there were some experimental evidences, such as XRD reflection peaks as well as the values of the microhardness and lattice parameter in the transformed and untransformed zones, which suggested the presence of a net flow of nitrogen from the untransformed N-supersaturated $\gamma_s$ zones to $\gamma$, along the precipitation process. Therefore, XPS and AES techniques could be used to establish final evidence of this phenomenon.

As it is possible to see from Figure 26a, the grains size of $\gamma_s$ is about 100 µm, while the dimension of the transformed areas is much smaller, at about 10 µm. A traditional XPS apparatus is not adequate for us to study the chemical composition of the transformed zones with sufficient resolution because it can analyze only surface areas between 0.1 and 1 mm. For this reason, it was necessary to use a scanning photoelectron microscopy (SPEM) operating in both imaging and spectroscopy modes. Indeed, this type of analysis can use an X-ray microprobe with a diameter less than 100 nm.

By using the SPEM technique, it was possible to determine the chemical composition and spatial distribution of the elements in the lamellae and interlamellar spaces. In Figure 27, a spatially resolved XPS image of the transformed zone before and after the topographical correction is shown.

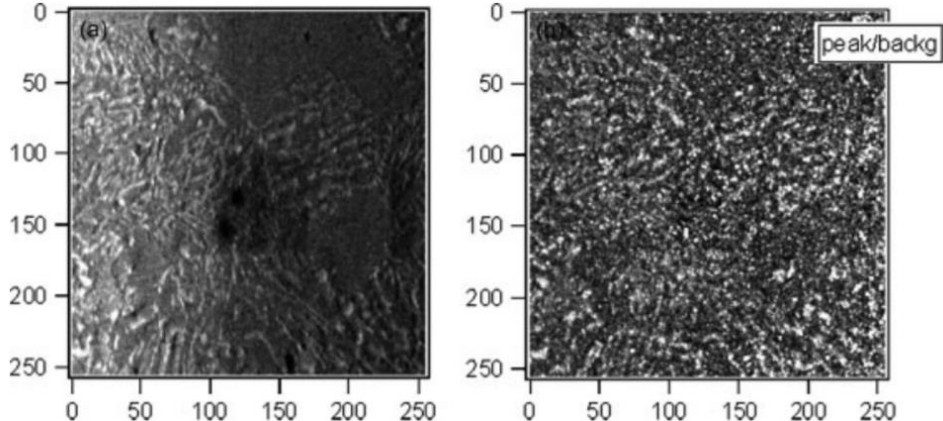

**Figure 27.** SPEM Cr 3p chemical images (51.2 × 51.2 μm²) of the sample treated for 23 h at 850 °C: (**a**) as-acquired and (**b**) corrected for the sample topography by calculating the ratio peak/background for Cr 3p signal [40].

The information obtained from these images and from the microscopy was in good agreement with traditional XPS measurements. From the SPEM images of Cr 3p signal, it was found that in the transformed zone, Cr is concentrated in the lamellae, whereas it is uniformly distributed in low concentration in the untransformed region. In an opposite way to Cr distribution, a Fe-enrichment in the untransformed zone and impoverishment in the lamellae were revealed from Fe 3p images. These analyses indicate a migration of Cr, which is mainly accumulated in the $Cr_2N$ precipitates across the interface between $\gamma$ and $\gamma_s$.

Furthermore, from the Auger spectra shown in Figure 28, the Cr/N atomic ratio was calculated. It was found to be equal to 2.9 and 5.9 for the transformed and untransformed zones, respectively. This result confirms the nitrogen enrichment in the transformed zones during the heat treatment. Moreover, another phenomenon was also explained. In fact, from these analyses it was possible to hypothesize that the precipitation of $Cr_2N$ takes place as long as the flow of nitrogen from the untransformed to the transformed area is present. Finally, when $\gamma$ and $\gamma_s$ zones have the same concentration of N, the precipitation process is stopped, even if not all the cells of the steel were transformed.

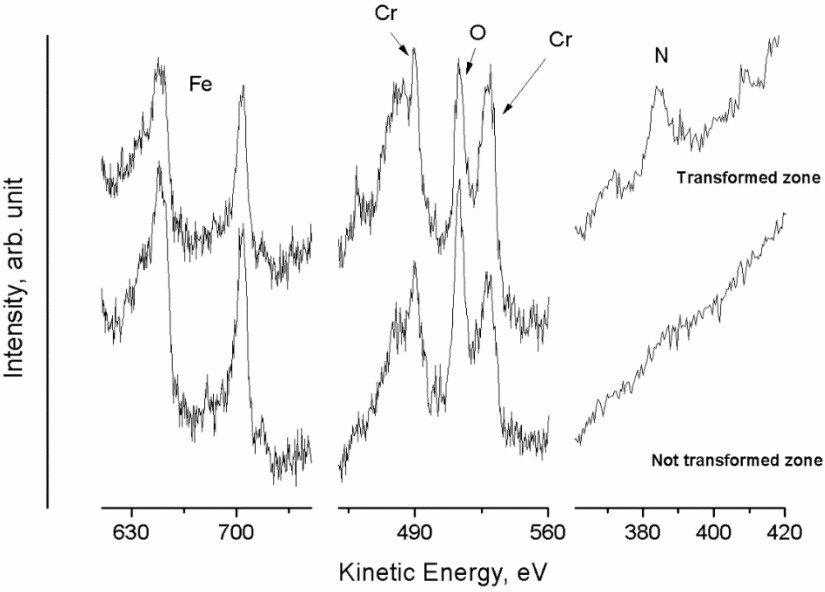

**Figure 28.** Auger spectra (Fe LMM, Cr LMM, O KLL, and N KLL) of untransformed and transformed zones of the steel sample after 23 h at 850 °C [40].

*3.7. Graphene on Polycrystalline Metals*

The last few decades of material science will be remembered as the years of the graphene revolution. In fact, although the theoretical predictions can be traced starting from the 19th century, the experimental evidences occurred only in the 2004. After that, throughout the scientific community there was a continuous race to discover the new fields of application in order to exploit the full potential of this 2D material. At the same time, it was essential to develop an industrial method of synthesis that could guarantee a large-scale production of graphene.

Recently, the research for the development of microelectronic devices, transparent conductive films, and in general different type of sensors with graphene has focused on the growth of graphene via the chemical vapor deposition (CVD) on polycrystalline metal substrates. These substrates act as excellent catalysts for the epitaxial growth of graphene. Some of them are also cheap and are easily removable, when it is necessary to transfer the layer of graphene on the device where it has to operate [42,43].

Numerous studies, extensively reviewed in [42–45], have been dedicated to the growth and characterization of graphene on various metals. Among many analytical techniques for the characterization of graphene, the mostly attractive ones are Raman spectroscopy, atomic force, and transmission electron microscopies, XPS and AES. However, in many papers including those on XPS, only the photoemission spectrum of C 1s has been used for graphene characterization, even if it does not allow the main peaks of graphene and graphite to be differentiated [44,46] without the angle-resolved analysis of low intensity $\sigma$ and $\pi$ bands accompanying the main peak [45]. A more useful and easier approach is the unequivocal identification of graphene from the analysis of C KVV spectrum combined with the main photoemission peaks of substrate and C 1s [46]. This approach, combined with Raman spectroscopy, allows us to obtain the information on the uniformity of graphene layer over a large area. In the same manner, these analyses permit us to determine the graphene thickness, which can often differ from the monolayer. The parameters, as well as the morphology and the thickness, depend on the type of growth mechanism of graphene. In the case of the CVD technique, two different growth mechanisms can take place: the decomposition of hydrocarbon gas at high temperature or the segregation of C atoms on the metal surface during the cooling phase.

For example, in the study in [47], the graphene was synthesized on the substrates of various polycrystalline metals. The growth was carried out by the CVD technique in a mixture of $CH_4$-$H_2$ gas at 1000 °C, with different times of exposure to the gas: 2, 4, and 6 min for the Cu substrate; and only 2 min for the Ni-Cu alloy (20 wt.% of Cu) and pure Ni film on Si substrate.

It was possible to make a preliminary test of graphene quality by Raman spectroscopy. At first, the disorder degree of the deposited films can be estimated from the intensity of the D-band (1350 cm$^{-1}$). Then, the ratio of the G-peaks band (1582 cm$^{-1}$) was calculated due to the presence of graphite or a multilayer system of graphene with respect to the typical signal of graphene G'-band (2700 cm$^{-1}$). The Raman spectra of graphene deposited on Cu foils are shown in Figure 29.

As it was observed from the value of the IG'/IG ratio, the sample exposed for 6 min to the gas mixture at 1000 °C appeared to be the most promising. This result was also confirmed by photoemission measurements.

Because it is not possible to distinguish between the graphite and graphene (both peaks are positioned at BE of about 284.5 eV) from the C 1s photoemission spectra, the Auger spectra of C KLL were also acquired. In fact, from the calculation of the D parameter, i.e., the distance between the absolute maximum and the absolute minimum of the first derivative of C KLL spectrum [39], it is possible to identify the presence of graphene [46]. Therefore, the value of D parameter was determined from the C KLL spectra induced by an X-ray source (XAES) and then it was compared with the same parameter obtained by using excitation with an electron gun (AES).

The typical spectra of C 1s and C KLL regions are shown in Figure 30, whereas all the results of the XPS characterization are summarized in the Tables 2–4.

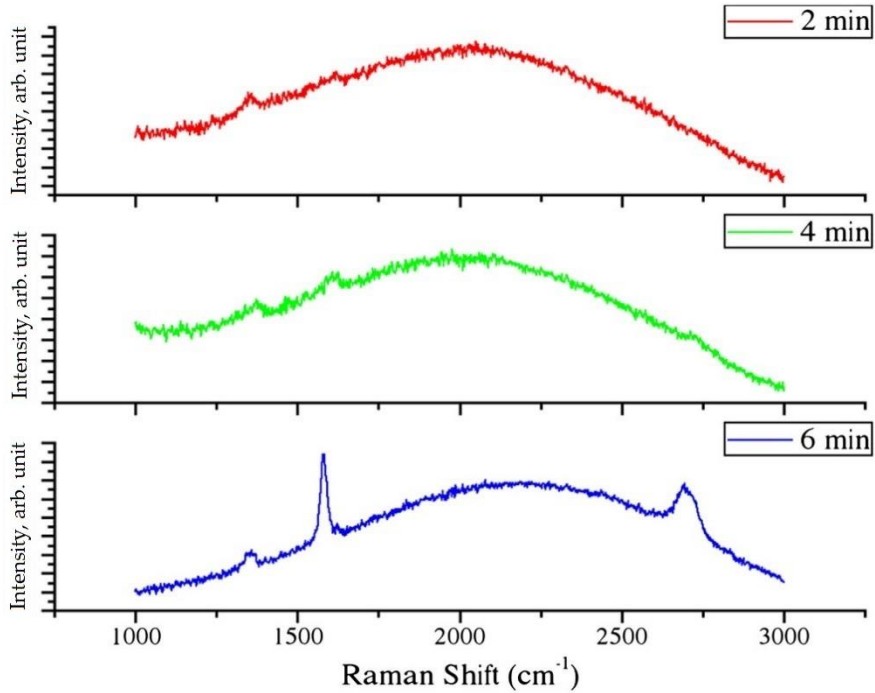

**Figure 29.** Raman spectra of graphene grown on electroformed Cu foils: exposure time of 2, 4, and 6 min [47].

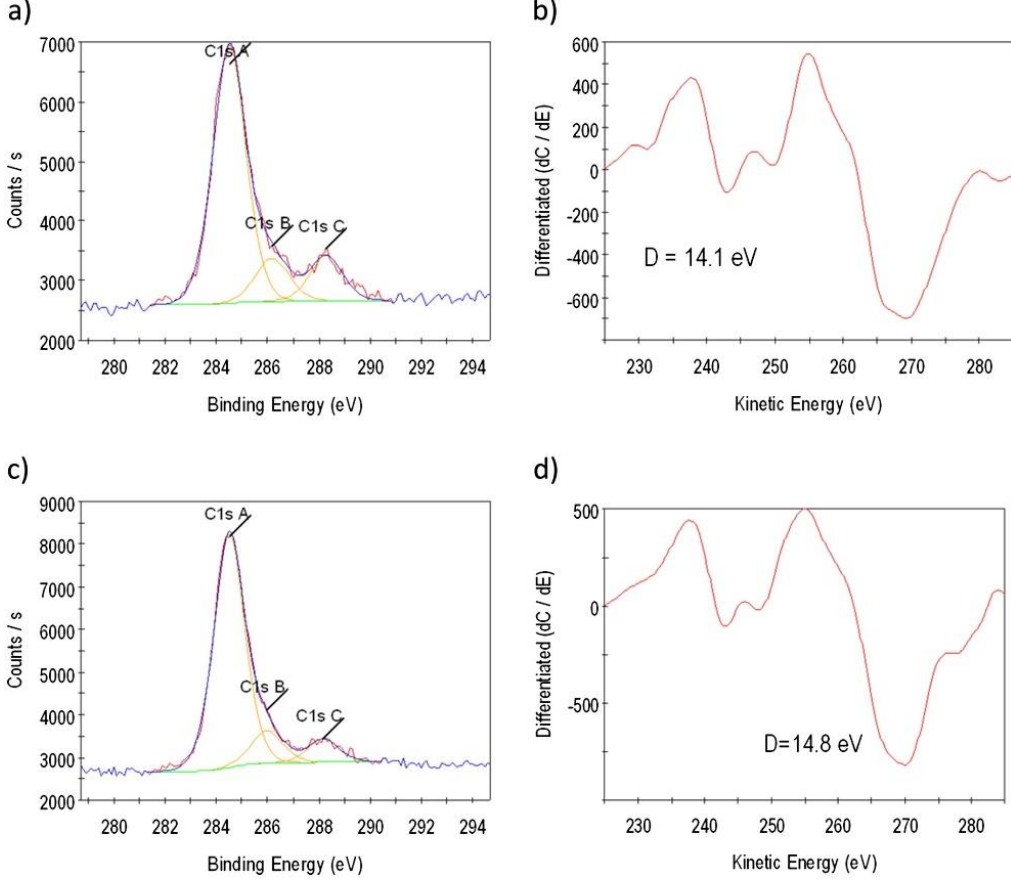

**Figure 30.** XPS spectra of C 1s for the samples 4 (**a**) and 6 (**c**) grown on Cu at 1000 °C. The corresponding C KLL spectra in the first derivative are shown in (**b**) and (**d**), respectively [47].

**Table 2.** Summary of XPS results for Sample 4 of graphene on Cu substrate [47].

| Sample 4 Peak | BE, eV | FWHM, eV | Atomic % | Bond |
|---|---|---|---|---|
| C1s A | 284.5 | 1.51 | 44.1 | C–C (graphene or graphite) |
| C1s B | 286.2 | 1.51 | 7.4 | C–O |
| C1s C | 288.3 | 1.51 | 7.7 | Carboxyl O–C=O |
| $Cu2p_{3/2}$ A | 932.5 | 1.71 | 9.3 | Cu(0) |
| $Cu2p_{3/2}$ B | 934.7 | 1.71 | 1.6 | $Cu(OH)_2$ |
| O1s A | 530.7 | 1.87 | 6.1 | C–O |
| O1s B | 531.8 | 1.87 | 20.9 | OH groups, C=O |
| Cl2p | 199.0 | 1.41 | 2.8 | - |

**Table 3.** Summary of XPS results for Sample 6 of graphene on Cu substrate [47].

| Sample 6 Peak | BE, eV | FWHM, eV | Atomic % | Bond |
|---|---|---|---|---|
| C1s A | 284.5 | 1.42 | 51.8 | C–C (graphene or graphite) |
| C1s B | 286.0 | 1.42 | 7.2 | C–O |
| C1s C | 288.2 | 1.42 | 5.2 | Carboxyl O–C=O |
| $Cu2p_{3/2}$ A | 932.4 | 1.51 | 12.8 | Cu(0) |
| $Cu2p_{3/2}$ B | 934.6 | 1.51 | 1.2 | $Cu(OH)_2$ |
| O1s A | 530.5 | 1.61 | 8.3 | C–O |
| O1s B | 531.8 | 1.61 | 12.3 | OH groups, C=O |
| Cl2p | 199.0 | 1.50 | 1.2 | - |

**Table 4.** D parameter (eV) determined by XPS and XAES for graphene samples on Cu substrate [47].

| Experimental | Sample 4 | Sample 6 | Description |
|---|---|---|---|
| XPS at 90° | 14.1 | 1.42 | Diamond-like |
| XPS at 45° | 13.3 | 14.2 | Diamond-like |
| XAES (e⁻ beam) | 22.1 | 21.5 | Graphitic |

From the Table 3, it is possible to conclude that the best graphene sample was obtained by the deposition of 6 min: The obtained values of the D parameter were $D_{XAES}$ = 14.1 eV (diamond-like) and $D_{AES}$ = 22.1 eV (graphitic) (see Table 4). As it was explained in detail in the previous work [46], these values definitely indicate the presence of graphene. From the XPS measurements at the grazing angle, it was also estimated that the thickness of graphene film was equal to a few monolayers.

In this way, a further example of the application of surface spectroscopic techniques demonstrated their versatility and potentiality in recent fields of scientific research and industrial development, such as the large-scale production of graphene.

## 4. Summary

The importance and potentiality of ESCA techniques for the exploration of metallic surfaces was illustrated by reviewing the main principles of these techniques and seven experimental cases of our research. The main techniques comprised in ESCA, i.e., X-ray photoemission and Auger electron spectroscopies, were successfully employed for the investigation of different metallic surfaces, and their modifications were induced by different treatments or operating conditions. In addition, the high resolution SPEM technique was applied for the exploration of submicrometric features of surface chemical composition in some of investigated materials.

Various phenomena on the metallic surfaces were revealed: the formation of impurity defects on collection coins, the microchemical composition and corrosion of stainless steel coated by Cr and Ti nitrides, modifications of microchemical composition in biphasic Ni-based superalloys, carbon diffusion at high temperature in the interface of Ti6Al4V/SiCf composite, microchemical inhomogeneity

of liquid PbBi alloy, surface modification of austenitic steels by plasma carburizing, and nitrogen migration at high temperature, with an influence of polycrystalline metal substrates (Cu, Ni, and NiCu alloy) on the growth of graphene. One more recent example of an advantageous ESCA application for the study of Cr segregation in martensitic stainless steel is reported in the present issue of this journal [48].

**Author Contributions:** Conceptualization, E.B., S.K., and A.M.; investigation, E.B., S.K., and A.M.; writing and editing, E.B., S.K., and A.M.; All authors have read and agreed to the published version of the manuscript.

**Funding:** No funding has been received for preparation of this review.

**Acknowledgments:** The authors are grateful to all the coauthors of reviewed papers for their contributions. Our special thanks are dedicated to Roberto Montanari (Tor Vergata University of Rome), who initiated the research in a major part of the reviewed cases, also to Luca Gregoratti and his team for hosting us at the ESCA microscopy beamline in the Elettra synchrotron and providing extensive experimental support.

**Conflicts of Interest:** The authors declare no conflict of interest.

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
