# Peer review of "ESCA as a Tool for Exploration of Metals’ Surface"

_coatings, doi:10.3390/coatings10121182_

Round 1

Reviewer 1 Report

"This article deals with the use of Auger electron spectroscopy and XPS photoelectron spectroscopy for chemical composition identification and chemical bond analysis in selected materials.

The results of the study presented in the work are valuable to any researcher, especially a beginner, who uses or plans to use these experimental methods in his or her research.

I find it particularly valuable to show the analytical capabilities of ESCA methods and the frequent problems that accompany ECSA surface analysis.

The authors explained the causes of the problems and how to solve them in concrete examples of experimental results of a wide spectrum of materials, such as noble metals, nitride coating on stainless steel, Ni- based superalloys and composites. Since AES and XPS are among the most commonly used in the analysis of materials’ surface, I think this article will find many readers and I recommend it for publication in the Coatings journal. The work is well written, richly illustrated.

The language of work is understandable. Minor editing errors were noted and in the transcript of the stoichiometry of the compounds. The part of the text highlighted as Conclusions in my opinion should be defined as a Summary".

(single text editing errors,
non-SI system pressure units used)

Author Response

We are sincerely grateful to both reviewers for valuable comments and suggestions. Our detailed answers are inserted below.

1) Comments and Suggestions for Authors

This article deals with the use of Auger electron spectroscopy and XPS photoelectron spectroscopy for chemical composition identification and chemical bond analysis in selected materials.

The results of the study presented in the work are valuable to any researcher, especially a beginner, who uses or plans to use these experimental methods in his or her research.

I find it particularly valuable to show the analytical capabilities of ESCA methods and the frequent problems that accompany ECSA surface analysis.

The authors explained the causes of the problems and how to solve them in concrete examples of experimental results of a wide spectrum of materials, such as noble metals, nitride coating on stainless steel, Ni- based superalloys and composites. Since AES and XPS are among the most commonly used in the analysis of materials’ surface, I think this article will find many readers and I recommend it for publication in the Coatings journal. The work is well written, richly illustrated.

The language of work is understandable. Minor editing errors were noted and in the transcript of the stoichiometry of the compounds. The part of the text highlighted as Conclusions in my opinion should be defined as a Summary.

Acknowledged and corrected.

(single text editing errors, non-SI system pressure units used)

We have used mbar unit for the pressure, as it is commonly accepted in the surface analysis and UHV community. Of course, if also the Editor thinks that only SI units must be used, we can easily change to Pa unit.

Reviewer 2 Report

The Authors provide a review of works dealing on the combination of XPS and Auger spectromicroscopies for investigating metal surfaces.

While putting together these results can certainly provide for a reference for those who are dealing with metal surfaces, I feel that the manuscript has to be revised in a way that it will put more emphasis on the techniques rather than a sort of historical view of the works from the authors. In its present form the reader has not a clear motivation to go through the different sections of the manuscript because the information is not introduced but rather comes across the reading. Since there is no novelty, an effort should be provided in structuring the results not in a self-centered list of papers but rather on a technique centered work.

Here are some points to help going in this direction.

The introduction should be more pedagogic, not a textbook but focused on metal surfaces. I suggest putting a figure illustrating energy levels and transitions associated to XPS and AES and a table where the key information on the lateral resolution, information depth, sample requirement, sensitivity (detection limit) are reported in a schematic way.

Change the name of section 3 to a more meaningful chapter

Section 3.1 could be changed to XPS studies of corrosion phenomena on gold surfaces; in this way the focus is on the technique and the reader has a reason to go through the paragraph. The challenges of studying metal corrosion should be introduced.

Section 3.2 when dealing with metal oxides, especially TiOx depth profiles should take into account the reduction induced by Ar+ sputtering. This was shown in tens of papers, however the authors don't spend a word on this. Furthermore, elemental ratios are certainly a great way to investigate diffusion processes, however, the authors seem to exclude phase segregation and inhomogeneity, specifically, if Cr/N ratio is 1, how this excludes the formation of Cr2N? For instance there can be half Cr metal phase and half Cr2N phase, so microscopic analysis should be coupled to this information to be reliable.

Section 3.3 investigation of phase separation seems to be the core of this section so this has to be introduced at the beginning of the paragraph.

The section of graphene is a little generic, the selection of papers is too self-centered considered the fact that much more accurate works are present in the literature of graphene

I could go through other details, however I think can go through their text keeping in mind my starting recommandations

Author Response

We are sincerely grateful to both reviewers for valuable comments and suggestions. Our detailed answers are inserted below.

2) Comments and Suggestions for Authors

The Authors provide a review of works dealing on the combination of XPS and Auger spectromicroscopies for investigating metal surfaces.

While putting together these results can certainly provide for a reference for those who are dealing with metal surfaces, I feel that the manuscript has to be revised in a way that it will put more emphasis on the techniques rather than a sort of historical view of the works from the authors. In its present form the reader has not a clear motivation to go through the different sections of the manuscript because the information is not introduced but rather comes across the reading. Since there is no novelty, an effort should be provided in structuring the results not in a self-centered list of papers but rather on a technique centered work.

Our aim was to illustrate the ESCA application for investigation of metals’ surface by experimental studies of various materials. As in our lab we had a lot of experimental cases (more than those overviewed in the present manuscript), we have selected the most interesting ones, seeking to illustrate specific applications for different materials. We agree that the approach of the “technique centered work” could be also useful, but our choice of the “material centered cases” has some advantages, e.g., the readers can skip some chapters and concentrate on the cases that are more interesting for their applications.

Here are some points to help going in this direction.

The introduction should be more pedagogic, not a textbook but focused on metal surfaces. I suggest putting a figure illustrating energy levels and transitions associated to XPS and AES and a table where the key information on the lateral resolution, information depth, sample requirement, sensitivity (detection limit) are reported in a schematic way.

Acknowledged and added.

Change the name of section 3 to a more meaningful chapter

Acknowledged and corrected.

Section 3.1 could be changed to XPS studies of corrosion phenomena on gold surfaces; in this way the focus is on the technique and the reader has a reason to go through the paragraph. The challenges of studying metal corrosion should be introduced.

Acknowledged and corrected.

Section 3.2 when dealing with metal oxides, especially TiOx depth profiles should take into account the reduction induced by Ar+ sputtering. This was shown in tens of papers, however the authors don't spend a word on this.

Acknowledged and corrected.

Furthermore, elemental ratios are certainly a great way to investigate diffusion processes, however, the authors seem to exclude phase segregation and inhomogeneity, specifically, if Cr/N ratio is 1, how this excludes the formation of Cr2N? For instance there can be half Cr metal phase and half Cr2N phase, so microscopic analysis should be coupled to this information to be reliable.

We have excluded the presence of Cr2N on the basis of Cr 2p3/2 and N 1s peak analysis: both of them indicated the presence of CrN. The spectrum of Cr 2p3/2 contained only a single component attributed to CrN, whereas the Cr2N phase is characterized by significantly higher value of BE (almost by 2 eV) [5]. The atomic ratio of Cr/N = 1 confirmed this supposition. This explanation was added to the text.

Section 3.3 investigation of phase separation seems to be the core of this section so this has to be introduced at the beginning of the paragraph.

Acknowledged and added.

The section of graphene is a little generic, the selection of papers is too self-centered considered the fact that much more accurate works are present in the literature of graphene.

Acknowledged and added.

I could go through other details, however I think can go through their text keeping in mind my starting recommandations

Round 2

Reviewer 2 Report

I am satisfied by the manuscript modifications and suggest the publication of the work in its present form. One detail, I personally do not recommend putting XPS peaks FWHM with two digits, as in Table 3, 4, I find more meaningful, considering fitting accuracy, to repor only one digit (i.e. 1.4 intead of 1.42 and so on).

Regards